# Persistent Roseoloviruses Infection in Adult Patients with Epilepsy

**DOI:** 10.3390/brainsci10050287

**Published:** 2020-05-11

**Authors:** Santa Rasa-Dzelzkaleja, Sabine Gravelsina, Svetlana Chapenko, Zaiga-Nora Krukle, Simons Svirskis, Normunds Suna, Elena Kashuba, Guntis Karelis, Modra Murovska

**Affiliations:** 1Institute of Microbiology and Virology, Rīga Stradiņš University, LV-1067 Rīga, Latvia; santa.rasa-dzelzkaleja@rsu.lv (S.R.-D.); scapenko@latnet.lv (S.C.); zaiga.nora@rsu.lv (Z.-N.K.); s.svirskis@latnet.lv (S.S.); modra.murovska@rsu.lv (M.M.); 2Department of Neurology and Neurosurgery, Riga East Clinical University Hospital “Gailezers”, LV-1038 Rīga, Latvia; n.suuna@gmail.com (N.S.); guntis.karelis@rsu.lv (G.K.); 3Department of Microbiology, Tumor and Cell Biology (MTC), Karolinska Institutet, 171 65 Solna, Sweden; elena.kashuba@ki.se

**Keywords:** human herpesvirus-6, human herpesvirus-7, reactivation, epilepsy

## Abstract

Background: Human herpesviruses (HHV)-6A, HHV-6B and HHV-7 are considered to be involved in the pathogenesis of epilepsy, a common neurological disorder. The objective of this study was to determine the association of roseoloviruses infection with epilepsy. Methods: 53 epilepsy patients and 104 ordinary blood donors were analyzed to determine presence of virus-specific antibodies by enzyme-linked immunosorbent assay (ELISA) and immunofluorescence assay (IFA), genomic sequences, viral load and gene expression by polymerase chain reactions (PCRs) and restriction analysis, HHV-6 protein expression by IFA and level of cytokines by ELISA. Results: Roseoloviruses genomic sequences in DNA samples from whole blood were found in 86.8% of patients versus 54.8% of controls and active infection was revealed only in patients with epilepsy (19.6% of roseolovirus-positive patients). Significantly higher viral load and more frequent gene expression was detected in patients compared to the controls. HHV-6-encoded protein expression was demonstrated in 53.3% of patients with previously detected HHV-6 DNA. Changes in level of cytokines were determined in patients with elevated viral load compared to the patients without elevated viral loads and to the controls. Conclusions: Results on frequent active HHV-6 and HHV-7 infection in epilepsy patient’ peripheral blood indicate on possible involvement of these viruses in the disease development.

## 1. Introduction

Epilepsy is a common chronic neurological disease characterized by recurring unprovoked seizures [1] due to abnormal excessive or synchronous neuronal activity in the brain. It is also characterized by the neurobiologic, cognitive, psychological and social consequences of this condition [2]. The disease can occur at any age and affects 1% of the human population worldwide [3]. Despite over 20 antiepileptic drugs being available, seizures remain uncontrolled in about 30% of patients [4] that can be related with the complex and multifactorial nature of epilepsy and its heterogeneity [5,6,7,8]. Viral infections of the CNS are among major risk factors for development of epilepsy [9].

Human herpesvirus (HHV)-6A, HHV-6B and HHV-7 belong to the Herpesviridae family, Betaherpesvirinae subfamily and *Roseolovirus* genus [10]. They are ubiquitous, lymphotropic, immunomodulating and potentially pathogenic to the nervous system [11,12].

The viruses enter in the brain via the olfactory pathways [13,14]. The process of the virus entry into the host cell is a complex interplay of multiple viral envelope proteins and cellular structures, prior studies [15], showed that both HHV-6A and B use CD46 as cell receptor. It has been observed that these viruses induce a cytokine imbalance with a switch from an antiviral T helper (Th)-1-polarized cytokine profile to a Th-2 profile [16,17] and significantly modulate the chemokine system both by affecting chemokine production and by expressing viral chemokines and chemokine receptors [18].

HHV-7 virus utilizes the CD4+ molecule as its essential receptor for entry into target cells [19] and induces the cytopathic effects in productively infected cells and apoptosis in uninfected or non-productively infected cells [20]. The routes of the virus entry into the CNS is unclear. Likewise, HHV-6, HHV-7 also can modulate the cytokine secretion from human PBMC and significantly modulate the chemokine system [21,22]. Primary infection with HHV-6B and HHV-7 occurs during infancy and can manifest as the febrile illness exanthema subitum (roseola) [23,24] which results in different symptoms or diseases, ranging from asymptomatic infection to acute febrile illness with severe neurological complications. Primary infection with HHV-6B may be the cause of approximately 30% of the febrile convulsions in children [25] and was associated with complications such as hemiplegia, meningoencephalitis and residual encephalopathy of primary infection [26,27]. Symptomatic HHV-6A primary infection is not that well documented. Some data suggest that HHV-6A may be more neurotropic and neurovirulent [28].

The pathogenic role of HHV-7 is not clear. HHV-7 primary infection has the potential for severe complications similar to HHV-6 [29,30,31,32,33] and may cause several clinical complications such as febrile seizures and encephalopathy in children, encephalitis and flaccid paralysis in immunocompetent [33] and immunocompromised adults.

The association of HHV-6B and HHV-7 primary infection with febrile status epilepticus has been underscored [30,34]. The viral primary infection proceeds with encephalitis and early seizures present an increased risk of late unprovoked seizures and epilepsy development [35,36,37,38].

After primary infection, the viruses establish a state of life-long sub-clinical persistence or latency in various sites and cells of the organism, including the CNS [39,40] and can be reactivated under a variety of stimuli especially in case of an immunodeficiency. The complications and encephalitis, in particular which is associated with reactivated roseoloviruses infection, has been observed in immunocompetent [41,42,43] as well as in immunocompromised adults [41,44] suggesting that the complications are due to viral invasion of the brain [12,32,40]. Late unprovoked epileptic seizures may be sequelae of viral encephalitis. The clinical manifestations show that seizures occurred more frequently in case of reactivated infection [45].

Controversial link of HHV-6B to the future development of intractable temporal lobe epilepsy and hippocampal sclerosis, in both children and adults have been shown [46,47,48,49,50,51,52,53]. A possible association between HHV-7 positivity and drug-resistant epilepsy, especially hippocampal sclerosis has been suggested [54], however, no association between HHV-7 and epilepsy has been confirmed. Although, the pathogenic potential of HHV-7 reactivated infection is unclear, however it can reactivate HHV-6 from latency and thus contributes to severe pathological conditions associated with HHV-6 [55,56].

HHV-6 and HHV-7 has been associated with neurological pathologies such as multiple sclerosis [57,58,59,60,61,62,63], myalgic encephalomyelitis/chronic fatigue syndrome [57,64,65,66,67], Alzheimer’s disease [68], as well as with Hashimoto’s thyroiditis [69,70]. Although a significant number of studies have suggested that the CNS can be a site for persistent HHV-6 and HHV-7 infection, the causal relationship and possible pathological role of these viruses in the human CNS disease and epilepsy are to be elucidated.

The objectives of this study were to (1) examine the presence of HHV-6 and/or HHV-7 DNA in whole peripheral blood (WPB) DNA samples from adult patients with different forms of epilepsy; (2) determine HHV-6A and HHV-6B; (3) compare HHV-6 and HHV-7 load in WPB DNA samples between the patients with epilepsy and apparently healthy controls; (4) reveal active roseoloviruses infection.

## 2. Materials and Methods

### 2.1. Samples

Peripheral blood samples of patients and healthy control group individuals were obtained during the period from 2014 to 2016. Clinical and virologic (including immunology and molecular biology) features were examined in the 53 patients having a confirmed clinical diagnosis of different forms of epilepsy (35 females and 18 males; median age 50, IQR 38–60; range 22–83) with various duration of epilepsy (1 year in 18.2%, 1–5 years in 25.5%, 6–10 years in 12.7%, 10–20 years in 23.6% and >20 years in 20.0%). As control group was used randomly selected 104 gender- and age-matched voluntary, apparently healthy blood donors (44 females and 60 males; median age 29, IQR 22–37; range 18–57). Ethylinediaminetetraacetic acid (EDTA) anticoagulated peripheral blood samples were collected once at State Blood Donor Center during ordinary blood donation at the same time period as patient samples. The patients with epilepsy were recruited at the Neurology Department of Riga East Clinical University Hospital “Gailezers” after admission with epileptic seizures. There was at least 24 h seizure-free period before recruitment of patients. Inclusion criteria for epilepsy group were as follows: (1) drug-resistant epilepsy, (2) status epilepticus on admission to hospital, (3) seizure frequency which is not typical for a patient (at least 100% increment in seizure frequency during the last month before admission). Exclusion criteria were as follows: (1) newly diagnosed epilepsy, (2) acute symptomatic seizure(s), (3) nonadherence to epilepsy treatment, (4) typical seizure in a patient with no evidence of drug resistance. Out of 53 epilepsy patients, the focal epilepsy was diagnosed in 85.0% (*n* = 45), genetic generalized epilepsy—in 5.6% (*n* = 3) and epilepsy of unknown type—in 9.4% (*n* = 5) of patients. Most common etiologies of epilepsy were: remote traumatic brain injury–in 26.4% (*n* = 14), cerebrovascular disease—in 20.7% (*n* = 11), cerebral palsy- in 13.2% (*n* = 7), postinfectious epilepsy- in 9.4% (*n* = 5), hippocampal sclerosis- in 5.7% (*n* = 3). Two patients had remote intracranial surgery due to anaplastic meningioma and meningosarcoma with no evidence of recurrence. One patient was diagnosed with autoimmune epilepsy caused by NMDA-receptor encephalitis, one- with focal cortical dysplasia and chronic alcohol-related brain damage was etiology of epilepsy in a single patient. Only one patient received immunosuppressive treatment with prednisone (Table 1). All of the patients received specific anti-epileptic treatment with monotherapy in 43.4% (*n* = 23) and polytherapy in the rest (56.6%, *n* = 30). Drug-resistant epilepsy was determined in 32.1% (*n* = 17) of the patients, others reported partial seizure control, and there were no patients experiencing seizure freedom within the last year before recruitment. Status epilepticus was a reason for hospital admission in 28.3% (*n* = 15). All subjects gave their informed consent for inclusion before they participated in the study. The study protocol was approved by Ethics Committee of the Rīga Stradiņš University (30.05.2013.) code- Nr 28 and written informed consent was obtained from all participants prior to the examination.

### 2.2. HHV-6 and HHV-7 Serology

Testing for HHV-6 and HHV-7 IgG class antibodies in blood plasma samples was carried out using HHV-6 IgG ELISA kit (Advanced Biotechnologies, Columbia, MD, USA) and HHV-7 using IgG IFA kit (Advanced Biotechnologies, Columbia, MD, USA) in accordance with the manufacturer’s recommendations.

### 2.3. Nested Polymerase Chain Reaction

Nested polymerase chain reaction (nPCR) was used to amplify specific virus DNA sequences in DNA isolated from WPB (a marker of virus persistence) and cell-free blood plasma (a marker of persistent infection in active phase). Total DNA was isolated from fresh WPB by phenol-chloroform extraction method. The QIAamp DNA Blood Kit (Qiagen, Hilden, Germany) was used to extract DNA from blood plasma. The fresh plasma samples were treated with deoxyribonuclease I before DNA purification. To assure the quality of the WPB DNA as well as exclude contamination of plasma DNA by cellular DNA, β-globin gene analysis was performed by PCR. nPCR amplification for the viruses was carried out in the presence of 1 µg of WPB DNA and 10 µL of plasma DNA (corresponding to 100 µL of plasma). HHV-6 and HHV-7 DNA were detected in accordance with Secchiero et al. (1995) [71] and Berneman et al. (1992) [10], respectively. Used primers were complementary to the gene that encodes major capsid proteins for both HHV-6A and HHV-6B and U10 gene for HHV-7 [72]. Positive controls (HHV-6 and HHV-7 genomic DNA; ABI, Columbia, MD, USA) and negative controls (DNA obtained from practically healthy HHV-6 and HHV-7 negative blood donors), as well as water controls were included in each experiment. The sensitivity of HHV-6-specific primers corresponded to three copies of the HHV-6 genome and the sensitivity of HHV-7-specific primers to one copy of the HHV-7 per reaction [63,73].

### 2.4. Restriction Endonuclease Analysis

Obtained nPCR amplification products were digested with HindIII restriction endonuclease (Thermo Scientific, Waltham, MA, USA) which cleaves HHV-6B 163 bp amplification product into 66 bp and 97 bp fragments, whereas does not cleave HHV-6A [71].

### 2.5. Reverse Transcription PCR assays

RNA was extracted from PBMC samples positive on HHV-6 and/or HHV-7 DNA of the patients with epilepsy and control individuals using TRI Reagent (Sigma-Aldrich, Saint Louis, MO, USA). Complementary DNA (cDNA) was synthesized with reverse transcription (RT) using commercially available Revert First Standard cDNA synthesis Kit (Thermo Scientific, Waltham, MA, USA) according to manufacturer´s recommendations. PCR was used to amplify virus specific DNA sequences in cDNA samples. HHV-6 U89/90 regulatory immediate-early or α gene expression was detected according to Van den Bosch et al. (2001) [74] using primers complementary to both HHV-6A and HHV-6B. HHV-7 U57 gene expression coding the major capsid protein was detected according to Ito et al. (2013) [75].

### 2.6. Quantitative Real-Time PCR

HHV-6 and HHV-7 loads were detected in all WPB samples positive for viral genomic sequences of the patients with epilepsy and control individuals using HHV-6 Real-TM Quant kit (Sacace, Biotechnologies, Como, Italy) and Realquality RQ-HHV-7 kit (AB ANALITICA, Padova, Italy) with BioRad CFX96 Real-time PCR System (BioRad, Hercules, CA, USA) according to the manufacturer’s recommendations.

### 2.7. HHV-6 Encoded Proteins Expression by Indirect Immunofluorescence

To determine HHV-6-encoded protein expression, indirect immunofluorescence assay (IFA) was performed on PBMC by specific monoclonal antibodies (MoAb) against HHV-6. PBMC were separated by Histopaque-1077 (Sigma-Aldrich, Saint Louis, MO, USA) gradient centrifugation. The cells were fixed in a mixture of cold methanol and acetone (1:1 at −20 °C) on microscope slides. Prior the staining, cells were rehydrated in phosphate buffer saline. The following primary mouse MoAb were used: anti-p41 (clone 6A5D12), HHV-6A and HHV-6B-specific, reacts in the nucleus of the cell; anti-gH (gp100) (clone OHV-3), HHV-6B specific and anti-gB (gp116) (clone OHV-1), specific for HHV-6A and HHV-6B, both associated with viral replication [76]. The antibodies were diluted 1:50 in the blocking buffer. Rabbit anti-mouse FITC-conjugated (Dako, Glostrup, Denmark) serum was used as secondary antibody. Hoechst 33,258 (Sigma-Aldrich, Saint Louis, MO, USA) was added in a concentration of 0.4 µg/mL to the secondary antibody for DNA staining. The images were captured using Eclipse 80i microscope with a cooled charge-coupled device camera (Nikon, Tokyo, Japan).

### 2.8. Cytokines Detection by Enzyme-Linked Immunosorbent Assay

Level of cytokines was analyzed with enzyme-linked immunosorbent assay (ELISA) in blood plasma samples, according to manufacturers’ protocols. TNF-α level was estimated using Biorbyt Human TNFα ELISA kit (Biorbyt, United Kingdom) with detection limit of <1 pg/mL. Level of IL-12 (p70) was detected with eBioscience Human IL-12p70 Platinium ELISA and IL-10–with eBioscience Human IL-10 Platinium ELISA (eBioscience Europe/International, Vienna, Austria) with detection limits–2.1 pg/mL and 1 pg/mL, respectively.

### 2.9. Statistical Analysis

All the graphs, calculations and statistical analyses were performed using GraphPad Prism software version 8.02 for Mac (GraphPad Software, San Diego, CA, USA). To test whether the collected numerical data are normally distributed, the D’Agostino and Pearson, Anderson–Darling and Shapiro–Wilk normality tests were applied. Homogeneity of variances was tested using F or Brown–Forsythe and Bartlett’s tests and, in a case of unequal SDs comparison of medians between different groups was performed using nonparametric Mann–Whitney (MW) test (for the analysis of HHV-6 and HHV-7 load) or Kruskal–Wallis (KW) test (for the analysis of plasma cytokine levels) followed by two-stage step-up method of Benjamini, Krieger and Yekutieli as post hoc procedure. Quantification limit (QL) of the viral load was set up as 10 viral DNS copies/10^6^ cells, for cytokines assessment—as 2 pg/mL. Viral load and cytokine levels below QL were uniformly set up as randomized values around (viral load) or above (cytokines) QL/2 [77]. Categorical data (detection frequency of HHV-6 and/or HHV-7 genomic sequences in the DNA isolated from WPB samples of the individuals with epilepsy and the control group) were assessed by Fisher’s exact (F) or chi-squared (Chi^2^) test. In order to better understand the differences between the groups and to better visualize them, the model of n equalization of subject groups was applied. For the characterization of the central tendency of variables and their distribution, median with interquartile range (IQR) was used. A two-tailed P-value less than 0.05 (*p* < 0.05) was considered as statistically significant for all tests.

## 3. Results

### 3.1. Seroepidemiology

Specific anti-HHV-6 IgG class antibodies were found in 81.1% (43/53) and specific anti-HHV-7 IgG class antibodies in 83.0% (44/53) of plasma samples from the patients with epilepsy. No difference in the presence of anti-HHV-6 and anti-HHV-7 IgG class antibodies between epilepsy patients and control group individuals (78.8%, 82/104 and 81.7%, 85/104, respectively) was detected (*p* = 1.0).

### 3.2. Presence of HHV-6 and/or HHV-7 Genomic Sequences

The presence of HHV-6 and/or HHV-7 genomic sequences in WPB and cell-free blood plasma DNA samples detected by nPCR is summarized in Table 1. Roseoloviruses genomic sequences in DNA samples from WPB were found in 86.8% (46/53) of patients with epilepsy versus 54.8% (57/104) in control group individuals (*p* < 0.0001). Although no difference in the detection rate of single HHV-6 and single HHV-7 DNAs between the groups was detected, the simultaneous presence of both virus DNA (co-infection) was slightly higher in WPB samples from the patients with epilepsy (*p* = 0.011). Plasma viremia (marker of infection active phase) was revealed only in patients with epilepsy (9/46, 19.6%) (Table 1).

HHV-6A was identified in two and HHV-6B in 13 out of 15 WPB samples from patients with epilepsy positive for HHV-6 genomic sequences. HHV-6B was identified in all HHV-6 positive DNA samples from control individuals. In reality in our case we are speaking about HHV-6 reflecting both HHV-6A and HHV-6B because in the study and control groups we have detected only two cases with HHV-6A.

### 3.3. HHV-6 U89/90 Gene Expression

Using RT-PCR, HHV-6 U89/90 immediate early gene expression in PBMC was found in 73.3% (11/15) of the patients with epilepsy with previously detected HHV-6 genomic sequence in WPB DNA samples by nPCR, between them also four patients with HHV-6 plasma viremia. HHV-6 U89/90 gene expression was not found in any of HHV-6 positive samples from the control individuals.

### 3.4. HHV-6 Load

Quantitative PCR analysis of HHV-6 in WPB DNA samples showed a statistically significant (*p* < 0.0001, Chi^2^) difference between the patients with epilepsy and control group individuals, when number of individuals with low loads (10 copies/10^6^ cells) and loads >10 copies/10^6^ cells were compared (Figure 1a).

The frequency of HHV-6 load >10 copies/10^6^ cells was detected in 66.7% (10/15) of the patients with epilepsy and in one out of 10 or 10% of the control group individuals (*p* < 0.0001, Chi^2^) both with positive results of viral DNA detection in WPB samples by nPCR (Figure 1b).

The median HHV-6 load in the patients with epilepsy was significantly higher than in control group individuals (174 (IQR: 7–699) and 5 (IQR: 4–6 copies/10^6^ cells, respectively, *p =* 0.0008, MW) (Figure 1b). Higher median load was detected in the WPB samples from patients with epilepsy with a persistent viral infection in an active phase (plasma viremia) in comparison to the patients with persistent infection in latent phase (1178 (IQR: 442–1777) and 389 (IQR: 115–630) copies/10^6^ cells, respectively, *p =* 0.016, MW). Median viral load >10 copies/10^6^ cells in WPB (672 (IQR: 174–1757)) was determined in 81.8% (9/11) of the patients with previously estimated HHV-6 U89/90 gene expression.

### 3.5. HHV-6-Encoded Proteins Expression

Using the IFA test HHV-6-encoded proteins expression was demonstrated in 53.3% (8/15) PBMC samples from the patients with epilepsy with previously detected HHV-6 genomic sequences in WPB DNA. Presence of HHV-6 p41, an early protein, encoded by the U27 gene was detected in one PBMC sample of the patient that was positive on HHV-6 U89/90 gene expression and with viral load <10 copies/10^6^ cells. Simultaneous presence of HHV-6 p41 and gH (gp100) (encoded by the U48 gene) proteins was observed in two samples and simultaneous p41, gH and gB (gp116) (encoded by U39) —in 5 of PBMC samples from the patients that were positive for U89/90 gene expression with median viral load 671.9 (IQR: 173.8–1757.0). Plasma viremia was detected in four out of eight (50.0%) patients with HHV-6-encoded proteins expression (two—p41+gH and two—p41+gH+gB).

### 3.6. HHV-7 U57 Gene Expression

Using RT-PCR, HHV-7 U57 gene expression, coding the late major capsid protein, was detected in 37.5% (15/40) of PBMC samples from patients with epilepsy positive for HHV-7 genomic sequences and in one out of 53 control samples (*p* = 0.0001, F). “From these 15 patients with U57 gene expression 12 (80%) were only with HHV-7 genomic sequences in WPB DNA, others 3 were with HHV-6 and HHV-7 specific sequences presented in WPB DNA samples.

### 3.7. HHV-7 Load

Significantly higher frequency of the HHV-7 load >10 copies/106 cells (22/40, 55.0% vs 14/53, 26.4%, respectively; *p* < 0.0001, Chi^2^; Figure 2a,b) as well as the median viral load (in a subjects with viral load >10 copies/106 cells) 502 (IQR: 193–2087) vs 174 (IQR: 113–248) copies/106 cells was found between the patients with epilepsy and control group individuals (*p* = 0.012, MW; Figure 2c).

Higher median load was detected in the WPB samples from patients with epilepsy with a persistent viral infection in an active phase (plasma viremia) in comparison with persistent infection in latent phase (1145 (IQR: 377–1530) and 323 (IQR: 154–414) copies/106 cells, respectively; *p* = 0.01, MW). In 9 out of 15 samples (60.0%) from the patients with epilepsy and in one control individual that were positive for HHV-7 U57 gene expression in PBMC, median HHV-7 load was 285 and 237 copies/106 cells, respectively. HHV-7 mRNA expression was detected also in 6 patients with epilepsy in whose WPB DNA samples virus specific sequences were presented with load <10 copies/10^6^ cells.

### 3.8. Level of Cytokines

Median level of TNF-α in the control group individuals without HHV-6A/B and HHV-7 infection markers was determined as 4.4 (IQR: 2.8–5.2) pg/mL. The median TNF-α level above QL (121.4, IQR: 66.5–205.5 pg/mL) in plasma samples from the patients with infection markers and elevated median HHV-7 load (>10 copies/10^6^ cells) was significantly higher (*p* = 0.0001, KW) in comparison to the patients without elevated HHV-7 load (<10 copies/10^6^ cells) (15.0, IQR: 10.6–27.5 pg/mL), as well as in comparison to control individuals without HHV infection (p< 0.0001, KW). Similar, but less pronounced difference (*p* = 0.042, KW) between both groups (<10 and >10 copies/106 cells) and in comparison to control (*p* = 0.0023, KW) was observed in a case of HHV-6 infection–median plasma TNF-α concentration was established at the level of 29.7 (IQR: 22.3–109.0) pg/mL in patients with elevated (>10 copies/10^6^ cells) HHV-6 load and at the level of 8.0 (IQR: 2.8–16.0) pg/mL in the patients without elevated HHV-6 load (<10 copies/10^6^ cells. Comparing groups with elevated (>10 copies/10^6^ cells) HHV-6 and HHV-7 viral load, in HHV-7 group significantly higher (*p* = 0.025, MW) median level of TNF-α was established (Figure 3a,b).

The median IL-12(p70) plasma level in samples from the patients with detectable viral infection markers (>10 copies/10^6^ cells) was similar in HHV-6 and HHV-7 groups (17.5, IQR: 14.9–24.8 and 16.1, IQR: 14.6–24.6, pg/mL, respectively), but both significantly differ (*p* = 0.019 and *p* = 0.032, respectively) from the control one (10.0, IQR: 9.8–12.1–pg).

Median levels of IL-12(p70) in the patients without elevated (<10) HHV-6 load (5.8, IQR: 3.9–7.9 pg/mL) was significantly lower in comparison to the group with elevated (>10) HHV-6 load (*p* = 0.0002, KW). In contrast, between patient groups with and without elevated HHV-7 load—(16.1, IQR: 14.6–24.6 and 10.0, IQR: 8.5–15.5 pg/mL, respectively) the difference was less pronounced (*p* = 0.048, KW) (Figure 4a,b).

Significant difference (*p* = 0.015, KW) between median levels of IL-10 in samples from epilepsy patients with (>10 copies/10^6^ cells) and without (<10 copies/106 cells) elevated HHV-7 load (34.5, IQR: 21.0–40.3 and 22.0, IQR: 8.5–30.5, respectively) was detected. Statistically significant difference of median IL-10 levels was also observed between control group individuals without roseoloviruses infection and patient group with HHV-7 infection (>10 copies/10^6^ cells) (15.5, IQR: 7.3–22.0 and 34.5, IQR: 21.0–40.3 pg/mL, respectively, *p* = 0.013, KW). In patients with elevated (>10) and without elevated HHV-6 load (<10 copies/10^6^ cells), the level of IL-10 was similar (19.3, IQR: 12.3–26.5 and 18.9, IQR: 6.7–37.8 pg/mL, respectively) and did not differ from control one. However, the median level of IL-10 was a little higher (*p* = 0.024, KW) in epilepsy patients with HHV-7 (>10 copies/10^6^ cells) in comparison to epilepsy patients with HHV-6 (>10 copies/10^6^ cells) (Figure 5a,b).

## 4. Discussion

Epilepsy is considered as a public health problem worldwide and is the most frequent neurological disorder. Viruses such as HHV-6A, HHV-6B and HHV-7 are hypothetical candidates for the involvement in epilepsy pathogenesis. Clarification of the relationship between causative role of HHV-6 and HHV-7 with chronic diseases of CNS, epilepsy in particular, is difficult due to the ubiquitous nature of the viruses, chronic persistent infection, two distinct species of HHV-6A and HHV-6B and limitations of current investigational tools that note many researchers [11,78]. The detection of active HHV-6 and/or HHV-7 infection is important for distinguishing between reactivation and latency of the viruses in patients with epilepsy.

In this study only HHV-6 and HHV-7 specific IgG class antibodies were measured. Taking into account the fact that primary (acute) infection with these viruses mainly occurs in early childhood and IgM antibodies appear only during an active infection or for 2–3 months after an active infection, serologic assays currently used for the detection of HHV-6 and HHV-7 specific IgM class antibodies have limited usefulness in the management of adult infection and especially in detection of active beta-herpesvirus infection. Therefore, use of methods such as RT-PCR to detect viral late genes expression and quantitative PCR to reveal viral load can help to receive more information on current status of the HHV-6/HHV-7 infection. It is known that the absence of an IgM antibody does not mean you do not have an active phase of persistent infection. Chronic infections in various tissues can persist with no evidence of IgM.

In this study, the presence of HHV-6 and HHV-7 IgG class antibodies in plasma samples from adult patients with epilepsy and control group individuals show high frequency of seroprevalence in both groups of individuals (81.1% and 83.0% vs 78.8% and 81.7%, respectively) that conforms to the data of other researchers [31,79].

The technique of nPCR was used for the detection of HHV-6 and HHV-7 genomic sequences in DNA isolated from WPB (marker of persistent infection) and cell-free blood plasma (marker of persistent infection in active phase). The results of this study show significantly higher frequency of a persistent roseoloviruses infection in the patients with epilepsy than in control group individuals without neurologic pathology (86.8% vs 54.8%, respectively; p = 0.003). The presence of HHV-6 DNA in WPB is detected significantly more often in the patients with epilepsy than controls (28.3% and 9.6%; *p* = 0.0048), similar effect was observed in a case of HHV-7 DNA presence in both groups (75.5% and 51%, respectively; *p* = 0.0035). Persistent roseolovirus infection in active phase is revealed only in patients with epilepsy (17.0%, *p* < 0.0001). The results confirm the possibility of this test to be used as one of the methods for diagnosis of active viral infection. However, negative findings of viral DNAs in the plasma of patients with epilepsy does not exclude active viral replication in different organs with no trace of HHV-6 and HHV-7 DNA in the plasma as usually these viruses have cell to cell spread tactic.

The comparative analysis of the data of quantitative HHV-6 PCR DNA test on WPB showed a significant difference between the means of HHV-6 load (*p* = 0.0008) and ratios of number of the individuals with load <10 and >10 copies/10^6^ cells (66.7% and 10.0%, respectively; *p* < 0.0001) in the patients with epilepsy and control group individuals (Figure 1a,b). In addition, IFA results of PBMC samples from the patients with epilepsy with elevated HHV-6 load in WPB also demonstrated expression of HHV-6 early (p41), glycoproteins gH(gp100) and gB(gp116) antigens in 63.6% of patients, indicating active viral replication. These results confirm that the quantitative method for determining viral load can be used to identify active viral infection. These results are in accordance with the data obtained by Donati et al. (2003) [46] on the surgical brain resection samples from patients with mesial temporal lobe epilepsy.

To confirm whether HHV-6 reactivation occurs in the patients with epilepsy, the detection of HHV-6 U89/90 gene expression in PBMC, HHV-6 load in WPB DNA, and the presence of HHV-6 encoded early (p41) and late virion envelope glycoproteins gH (gp100) and gB (gp116) expression in PBMC was done. Our results show that HHV-6 U89/90 regulatory immediate early or α gene expression in PBMC, which is believed to play an important role in virus latency and reactivation, are found only in the patients with epilepsy, but not in samples from the control individuals both with previously detected HHV-6 genomic sequence in WPB DNA by nPCR. This means that HHV-6 U89/90 gene transcript play an important role in the viral replication cycle and may be expressed not only during persistent infection in latent phase, but also during persistent infection in active phase, which coincides with the other researchers’ data [40,74,80]. However, additional study should be applied to make general conclusions. In 86.7% of the WPB samples from epilepsy patients and in all the samples from control group individuals HHV-6B was detected, indicating that either HHV-6A occurs infrequently in these individuals or that there is a difference in HHV-6A and HHV-6B geographical distribution, and HHV-6B in our area is predominant [39,66].

In our study we have used the HHV-6 IgG ELISA kit not allowing distinguish HHV-6A and HHV-6B which could be a possible reason for the insignificant difference in serology levels that does not agree with the results of the nested PCR. There are methods available now that can be used to distinguish between HHV-6A and B (IFA from Biocell Diagnostics, for example). The pathogenic role of HHV-7 in adult patients with epilepsy is not clear. A possible association between HHV-7 positivity and drug-resistant epilepsy, especially hippocampal sclerosis has been suggested, however no association between HHV-7 and epilepsy has been confirmed [54].

According to our study, the detection frequency of HHV-7 DNA in WPB significantly differs between the control group individuals (75.5% and 51.0%, respectively; *p* = 0.0035), however only in 15.0% of patients’ plasma viremia is revealed. Elevated viral load has 55.0% of the patients and 26.4% of control group individuals (*p* = 0.0094).

HHV-7 U57 gene expression, coding the major capsid protein, is detected significantly often among patients with epilepsy than control group individuals (*p* < 0.0001). From patients with positive results of this gene expression, single HHV-7 infection is diagnosed in most (80.0%) of the patients and half from them have elevated viral load. These results show higher activity rate of HHV-7 in patients with epilepsy-peripheral blood and thus it is indicating on HHV-7 possible involvement in the disease development, however, to make final conclusions additional research is required. The ability of HHV-7 to persist in brain tissues from encephalopathy patients’ autopsies has been previously demonstrated by our study group [39]; whenever herpesviruses persist in solid tissues, virions are rarely present in blood stream. It could be at least partial explanation for the absence of significant differences in nPCR results between patients and control group. Therefore, investigation of brain tissues for more detailed study of HHV-7 possible involvement in epilepsy development is required.

Pro-inflammatory cytokine TNF-α is produced following activation of nuclear factor–kappa B that controls inflammatory genes. Inflammation is stated to be involved in epilepsy. Meta-analysis reveals many studies of elevated TNF-α level in patients with epilepsy in serum and cerebrospinal fluid (CSF), however not in all reports [81].

In this study, significantly higher median TNF-α level is determined in patients with elevated HHV-7 load (>10 copies/10^6^ cells) comparing to the patients without elevated HHV-7 load (*p* < 0.0001), as well as to patients with elevated load of HHV-6 (*p* = 0.02) and patients without roseoloviruses infection (*p* < 0.0001). In addition, level of TNF-α is higher in patients with elevated HHV-6 load than without it (*p* = 0.028), also in comparison with patients without infection (*p* = 0.019). Among worldwide research, higher expression of TNF-α in CD4+ lymphocytes is reported in patients with temporal lobe epilepsy than controls [82]. In addition, elevated level of TNF-α is revealed in mice models for epilepsy [83]. However, some investigators find no significant changes of TNF-α level within various subgroups of patients and controls [16,84].

Elevated level of IL-12 is observed in CSF of patients with epilepsy [81]. Moreover, level of IL-12 is reported to be significantly increased in serum samples from experimental mice models of temporal lobe epilepsy compared to control group [83]. Current study shows significant difference in median levels of IL-12(p70) between the patients with elevated HHV-6 and HHV-7 viral load and without roseolovirus infection (*p* = 0.027 and *p* = 0.04, respectively), as well as the patients with and without elevated viral load of HHV-6 and HHV-7 (*p* = 0.0021 and *p* = 0.048, respectively). A bit higher level of IL-12(p70) have patients without roseolovirus infection than with HHV-6 <10 copies/10^6^ cells.

More often pro-inflammatory cytokines are found to be elevated in patients with epilepsy, however some researchers report increase of such anti-inflammatory cytokine as IL-10 in patients CSF. Patients with temporal lobe epilepsy have more IL-10 producing cells than control group individuals, and further immune cells show effector T-cells activation with low-grade inflammation [82]. In this study, a bit higher median level of IL-10 is revealed in epilepsy patients with elevated HHV-7 load comparing to the patients without elevated HHV-7 load (*p* = 0.015), elevated HHV-6 load (*p* = 0.024) and without roseoloviruses infection (*p* = 0.013). Statistically proven higher level of IL-10 in CSF is reported by several published studies elsewhere [81].

Despite a great deal of effort to understand whether a viral infection led to epilepsy or whether epilepsy caused viral activation, thus further deepening the clinical course of the disease, is challenging and unlikely to answer at this time, as further study of viral pathomechanisms and interplay with the disease is needed. Knowing the fact that primary infection with HHV-6B and HHV-7 occurs during infancy and can manifest as the febrile illness exanthema subitum (roseola) which results in different symptoms or diseases, ranging from asymptomatic infection to acute febrile illness with severe neurological complications and that the median age for acquisition of HHV-6 is nine months and 26 months for HHV-7, corresponding to the peak incidence of febrile seizures and febrile status epilepticus, and that the HHV-6 A, HHV-6B and HHV-7 are lymphotropic immunomodulating and potentially pathogenic to the nervous system, one of the speculations could be that HHV-6 and/or HHV-7 are among the infectious agents that can provoke development of epilepsy. For sure that further studies should aim at completing the picture on the interplay among those viruses and for sure the groups should be larger.

## 5. Conclusions

Results on frequent active HHV-6 and HHV-7 infection in patients with epilepsy peripheral blood indicate on possible involvement of these viruses in the disease development, however, more evidences are required, such as data on viral presence in tissue from specific regions of the brain.

## Figures and Tables

**Figure 1 brainsci-10-00287-f001:**
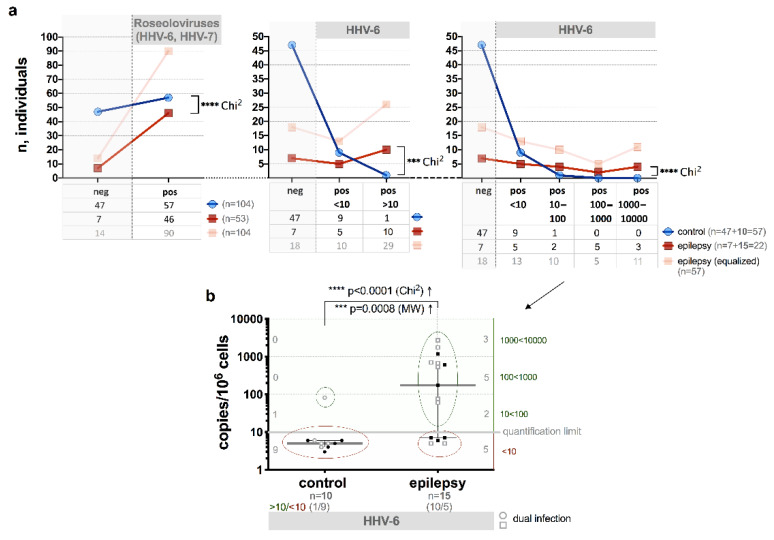
HHV-6 load in whole peripheral blood (WPB) of the patients with epilepsy (including cases with dual HHV-6 and HHV-7 infection) and control group individuals: (**a**) Comparison of frequency of HHV-6 positive (pos) and negative (neg) cases in epilepsy patients and control individuals; (**b**) comparison of viral load and frequency of HHV-6 positive epilepsy patients and control individuals, light symbols indicate cases with double HHV-6 and HHV-7 infection, with red ovals are marked low viral load (<10 copies/10^6^ cells) cases, with green ovals—elevated viral load (>10 copies/10^6^ cells) cases; asterisks in **a** represent a significance level of differences between groups (*** *p* < 0.001, **** *p* < 0.0001).

**Figure 2 brainsci-10-00287-f002:**
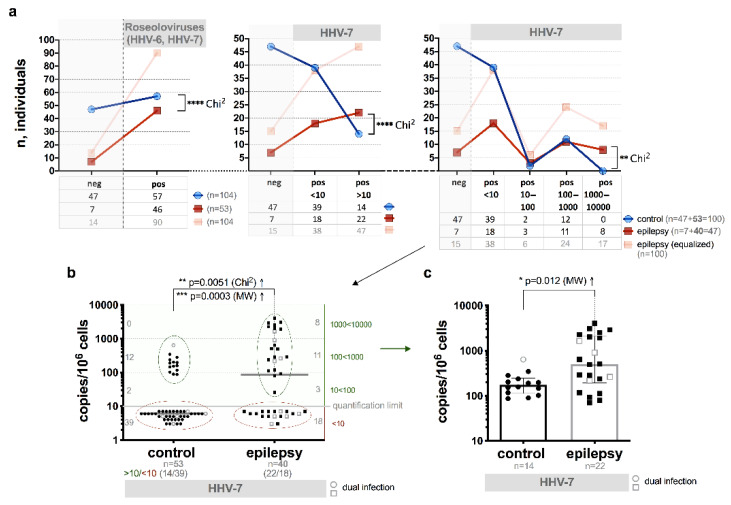
HHV-7 load in whole peripheral blood (WPB) of the patients with epilepsy (including cases with dual HHV-7 and HHV-6 infection) and control group individuals: (**a**) comparison of frequency of HHV-7 positive (pos) and negative (neg) cases in epilepsy patients and control individuals; (**b**) comparison of viral load and frequency of HHV-7 positive epilepsy patients and control individuals, with red ovals are marked low viral load (<10 copies/10^6^ cells) cases, with green ovals—elevated viral load (> 10 copies/10^6^ cells) cases; (**c**) comparison of elevated viral loads of HHV-7 in epilepsy patients and control individuals; light symbols indicate cases with double HHV-7 and HHV-6 infection (**b,c**); asterisks in **a** represent a significance level of differences between groups (** *p* < 0.01, **** *p* < 0.0001).

**Figure 3 brainsci-10-00287-f003:**
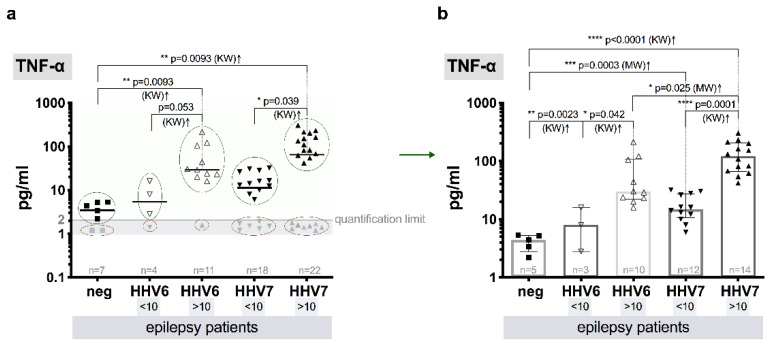
Comparative blood plasma level of TNF-α in epilepsy patients: (**a**) scatter plots with medians of HHV-6 and HHV-7 negative (neg) and HHV-6 and HHV-7 positive viral load (<10 and >10 copies/106 cells), gray symbols (marked with red ovals) show values below quantification limit (QL); (**b**) scatter plots with bars (medians with IQR) that represent values above QL (marked with green ovals in a) of HHV-6 and HHV-7 negative (neg) and HHV-6 and HHV-7 positive viral load (<10 and >10 copies/10^6^ cells), values below QL are excluded; asterisks and respective p value represent significance level of differences between the groups (KW—Kruskal–Wallis test).

**Figure 4 brainsci-10-00287-f004:**
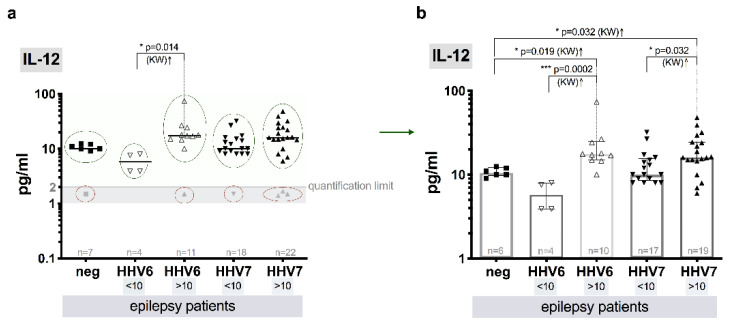
Comparative blood plasma level of IL-12 in epilepsy patients: (**a**) scatter plots with medians of HHV-6 and HHV-7 negative (neg) and HHV-6 and HHV-7 positive viral load (<10 and >10 copies/10^6^ cells), gray symbols (marked with red ovals) show values below quantification limit (QL); (**b**) scatter plots with bars (medians with IQR) that represent values above QL (marked with green ovals in a) of HHV-6 and HHV-7 negative (neg) and HHV-6 and HHV-7 positive viral load (<10 and >10 copies/10^6^ cells), values below QL are excluded; asterisks and respective p represent significance level of differences between the groups (KW—Kruskal–Wallis test).

**Figure 5 brainsci-10-00287-f005:**
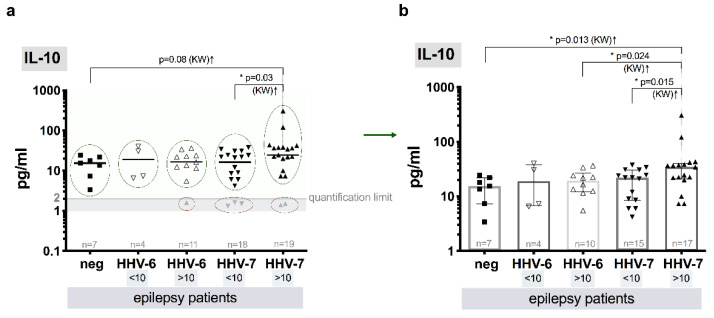
Comparative blood plasma level of IL-10 in epilepsy patients: (**a**) scatter plots with medians of HHV-6 and HHV-7 negative (neg) and HHV-6 and HHV-7 positive viral load (<10 and >10 copies/10^6^ cells), gray symbols (marked with red ovals) show values below quantification limit (QL); (**b**) scatter plots with bars (medians with IQR) that represent values above QL (marked with green ovals in a) of HHV-6 and HHV-7 negative (neg) and HHV-6 and HHV-7 positive viral load (<10 and >10 copies/10^6^ cells), values below QL are excluded; asterisks and respective p represent significance level of differences between the groups (KW—Kruskal–Wallis test).

**Table 1 brainsci-10-00287-t001:** Presence of HHV-6 and/or HHV-7 genomic sequences in whole peripheral blood and cell-free plasma DNA samples of the patients with epilepsy and control group individuals.

Samples	Viral Genomic Sequences						Total Positive
	Single HHV-6		Single HHV-7		Co-Infection HHV-6 + HHV-7		
	WPB	Plasma	WPB	Plasma	WPB	Plasma	
Epilepsy *n* = 53	6 (11.3%)	2 ^b^ (33.3%)	31 (58.5%)	5 ^b^ (16.1%)	9 * (17.0%)	2 ^a,b^ (22.2%)	46 ** (86.8%)
Control group *n* = 104	4 (3.8%)	0	47 (45.2%)	0	6 (5.8%)	0	57 (54.8%)

HHV-6—human herpesvirus-6, HHV-7—human herpesvirus-7, nPCR—nested polymerase chain reaction, WPB—whole peripheral blood, *n*—number of individuals; significance levels vs. control—* *p* = 0.011, ** *p* = 0.003, Fisher’s exact test; ^a^ HHV-6 genomic sequence only in one and both viruses genomic sequences in second sample were detected, ^b^ number of individuals from WPB positives.

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
