# Peer review of "Persistent Roseoloviruses Infection in Adult Patients with Epilepsy"

_brainsci, 2020, doi:10.3390/brainsci10050287_

Round 1

Reviewer 1 Report

This manuscript evaluated the association between reseoloviruses infection and epilepsy. Because there are numerous preclinical and clinical datasets that suggest that viral infections and/or encephalitis can contribute to seizure induction and maintenance, as well as be a causative factor for acquired epilepsy, this study is important to undertake to better define the association between common these common viruses and epilepsy. The methodology is standard in the field, the approach clear, and the interpretation reasonable. I have only minor comments to provide the authors to improve the clarity and impact of this manuscript.

  • The article does not really define whether their findings are causative of epilepsy, or instead an effect of the epilepsy condition itself. Because the patient population was quite heterogenous, a clear cause-effect relationship is challenging to interpret. Is the immune system of patients with epilepsy less able to clear an active or persistent infection; or does the infection itself lead an individual who is predisposed to develop epilepsy to become more susceptible to developing epilepsy? This should be further discussed.
  • The patient population is rather imbalanced with regards to gender. In the case population, there are more females than males (35:18) whereas in the control cohort, the balance is skewed towards males (44:60). While there is limited reason to believe that epilepsy in females is more severe or leads to challenges that may confound interpretation, there is limited evidence to suggest that the immune response in males vs females is different, which may be worth considering in the authors’ interpretation. See for examples: https://doi.org/10.1038/nri.2016.90; 3389/fimmu.2018.01931. The effect of sex as a biological variable therefore should be further discussed in the study and I would suggest reviewing the dataset to also include sex differences in analysis.
  • There were n = 5 post-infectious epilepsy cases in the study group. Is it known what infectious agent contributed to the epilepsy in these patients? Was this group excluded from any post-hoc analysis?
  • The discussion is rather long and verbose. It might be helpful to reduce the length of the discussion to focus only on the most salient points of this study’s findings.
  • Some general grammatical review is suggested to improve clarity and overall readability.

Author Response

Dear Reviewer,

Thank you very much for the careful reviewing of our manuscript.  Please find the answers to your comments below:

  1. The article does not really define whether their findings are causative of epilepsy, or instead an effect of the epilepsy condition itself. Because the patient population was quite heterogenous, a clear cause-effect relationship is challenging to interpret. Is the immune system of patients with epilepsy less able to clear an active or persistent infection; or does the infection itself lead an individual who is predisposed to develop epilepsy to become more susceptible to developing epilepsy? This should be further discussed.

OUR ANSWER:

(Line 469-481)

Giving a specific answer to the question of whether a viral infection led to epilepsy or whether epilepsy caused viral activation, thus further deepening the clinical course of the disease, is challenging and unlikely to answer at this time, as further study of viral pathomechanisms and interplay with the disease is needed. Knowing the fact that primary infection with HHV-6B and HHV-7 occurs during infancy and can manifest as the febrile illness exanthema subitum (roseola) which results in different symptoms or diseases, ranging from asymptomatic infection to acute febrile illness with severe neurological complications and that the median age for acquisition of HHV-6 is 9 months and 26 months for HHV-7, corresponding to the peak incidence of febrile seizures and febrile status epilepticus, and that the HHV-6 A, HHV-6B and HHV-7 are lymphotropic immunomodulating and potentially pathogenic to the nervous system, one of the speculations could be that HHV-6 and/or HHV-7 are among the infectious agents that can provoke development of epilepsy. For sure that futher studies should aim at completing the picture on the interplay among those viruses and for sure the groups should be larger.

  1. The patient population is rather imbalanced with regards to gender. In the case population, there are more females than males (35:18) whereas in the control cohort, the balance is skewed towards males (44:60). While there is limited reason to believe that epilepsy in females is more severe or leads to challenges that may confound interpretation, there is limited evidence to suggest that the immune response in males vs females is different, which may be worth considering in the authors’ interpretation. See for examples: https://doi.org/10.1038/nri.2016.90; 3389/fimmu.2018.01931. The effect of sex as a biological variable therefore should be further discussed in the study and I would suggest reviewing the dataset to also include sex differences in analysis.

OUR ANSWER:

Based on the already known facts about HHV-6 and HHV-7, it is known that there is no relationship between genger and frequency of viral infection presence. There are no significant differences in the etiopathogenesis of epilepsy between women and men. However, we completely agree to you that question regarding gender and immune response is worth discussion and it cannot be ruled out that the immune response elicited by active viral infection may differ and this aspect should be further investigated. To complete the picture we should enlarge groups.

  1. There were n = 5 post-infectious epilepsy cases in the study group. Is it known what infectious agent contributed to the epilepsy in these patients? Was this group excluded from any post-hoc analysis?

OUR ANSWER:

The most frequently described etiological factors that caused epilepsy in the respective patients were evaluated and patients with post-infectious epilepsy were not included in the calculations.

Reviewer 2 Report

In this article, the authors address an important question of the role of human herpesviruses in the pathogenesis of epilepsy. The study benefits from being a controlled study, with large groups of epilepsy patients and control healthy subjects. The methods for assessing presence of antibodies, viral DNA and cytokines are sound.

The major limitation of this paper lies in the scant information provided about how and where the control patients were recruited. After all, any difference in exposure to the herpesviruses between the groups could be attributed to other differences in the groups themselves; were the controls taken from outpatients, while all of the epilepsy patients were recruited from an inpatient Epilepsy Monitoring Unit? Were the controls taken from a different hospital?

As of right now, the only information given about the controls is in this sentence on lines 100-102:

“…and randomly selected 104 gender- and age-matched apparently healthy control group individuals (44 females and 60 males; 102 median age 29, IQR 22 – 37; range 18 – 57). “

How the controls were selected is a crucial piece of information, and without knowing, it is difficult to draw any conclusions from this study.

If the controls were selected from inpatients from the same hospital during the same time period, then, please give us some idea of what they were admitted for. If the controls were selected from an outpatient setting, or a different hospital, then, this would need to be stated in the methods, addressed in the discussion, and mentioned in the abstract as well.

Minor issue:

Abstract

Line 14- “in pathogenesis of frequent neurological disorder – epilepsy” should be “in pathogenesis of epilepsy, a common neurological disorder” or “in pathogenesis of a common neurological disorder, epilepsy.”

Author Response

Dear Reviewer,

Thank you very much for the careful reviewing of our manuscript.  Please find the answers to your comments below:

Randomly selected 104 individuals who were used in this study as control group were not inpatients or outpatients, they were practically healthy blood donors. EDTA anticoagulated peripheral blood samples were collected once at Blood Donor Centre during ordinary blood donation at the same time period as patient samples .

We also made changes in the manuscript – added missing information about our control group (Line 102-106).